

# An Approach to Minimize Aircraft Motion Bias in Multi-Hole Probe Wind Measurements made by Small Unmanned Aerial Systems

Loiy Al-Ghussain[1] and Sean C. C. Bailey[1]

[1]Department of Mechanical Engineering, University of Kentucky, Lexington, Kentucky, USA 40506

**Correspondence:** Sean Bailey (sean.bailey@uky.edu)

**Abstract.** Multi-hole probe mounted on an aircraft provide the air velocity vector relative to the aircraft, requiring knowledge of the aircraft spatial orientation, translation, and velocity to translate this information to an Earth-based reference frame and determine the wind vector. As the relative velocity of the aircraft is typically an order of magnitude higher than the wind velocity, the extracted wind velocity is very sensitive to multiple sources of error including misalignment of the probe and aircraft coordinate system axes, sensor error and misalignment in time of the probe and aircraft orientation measurements in addition to aerodynamic distortion of the velocity field by the aircraft. Here, we present an approach which can be applied after a flight to identify and correct biases which may be introduced into the final wind measurement. The approach was validated using a ground reference, different aircraft, and for the same aircraft at different times. The results indicate significant reduction in wind velocity variance at frequencies which correspond to aircraft motion.

## 1 Introduction

The past few decades have witnessed significant increase in the utilization of small unmanned aerial systems (sUAS) in wide range of atmospheric research areas such as the evolution and structure of the atmospheric boundary layer (see, for example, van den Kroonenberg et al., 2007; Van den Kroonenberg et al., 2008; Cassano et al., 2010; Bonin et al., 2013; Lothon et al., 2014; Wildmann et al., 2015; Bärfuss et al., 2018; de Boer et al., 2018; Kral et al., 2018; Bailey et al., 2019), turbulence (Balsley et al., 2013; Witte et al., 2017; Bailey et al., 2019), analysis of aerosols and gas concentration in the atmosphere (Bärfuss et al., 2018; Platis et al., 2016; Corrigan et al., 2008; Schuyler and Guzman, 2017; Illingworth et al., 2014; Zhou et al., 2018), cloud microphysics (Ramanathan et al., 2007; Roberts et al., 2008) and observation and analysis of extreme weather events such as hurricanes (Cione et al., 2016). This increasing interest in sUAS is motivated in part by the rapid increase in their commercial development and use combined with advantages of sUAS over traditional ground-based measurement systems utilizing either remote sensing or in situ approaches. Specifically, sUAS can acquire high spatial and temporal resolution measurements in a relatively low-cost package that provides flexibility in measurement location and profile. In addition, when compared to manned aircraft measurements, their operation mitigates risks associated with measurement at lower altitudes and during



hazardous conditions or events (Elston et al., 2015; Bärfuss et al., 2018; Barbieri et al., 2019) such as erupted volcanoes with
ash covers (Pieri et al., 2017), near thunderstorms (Elston et al., 2011) and over contaminated regions (Bärfuss et al., 2018).

The most common atmospheric properties sampled using sUAS are pressure, humidity and wind (e.g. Egger et al., 2002; Hobbs et al., 2002; Balsley et al., 2013; Witte et al., 2017; Bärfuss et al., 2018; Rautenberg et al., 2018; Jacob et al., 2018; Barbieri et al., 2019; Bailey et al., 2019). Although these scalar quantities are relatively straightforward to acquire, obtaining all three components of the wind velocity vector is complicated by the presence of the continual translation and rotation of
the measurement platform, resulting in different approaches developed to determine the wind vector (Rautenberg et al., 2018; Suomi and Vihma, 2018; Laurence and Argrow, 2018; Shevchenko et al., 2016). Wind velocity measurements typically can be partitioned into approaches which: employ an on-board wind sensor and subtract the aircraft kinematics; employ the difference between the aircraft's relative air speed and ground speed (Suomi and Vihma, 2018; Cassano et al., 2016); using both techniques (Rautenberg et al., 2018); or through calibration of the aircraft's kinmatic and dynamic response to the wind (González-Rocha
et al., 2020).

Broadly speaking, sensor-based wind measurements tend to have higher temporal (and hence spatial) response, with sensors including sonic anemometers, single- and multi-hole pressure probes, and hot-wires (Suomi and Vihma, 2018). Usually, wind velocity measurements by fixed-wing sUAS require velocity probes with slightly better temporal response than sonic anemometers (Witte et al., 2017; Mayer et al., 2012). As a result, multi-hole pressure probes have been frequently used for
sUAS-based wind velocity measurements (Van den Kroonenberg et al., 2008; Elston et al., 2015; Spiess et al., 2007; Thomas et al., 2012) due to their high sampling frequency, light weight, simplicity, accuracy and almost linear relation between pressure and velocity at large wind velocities (Suomi and Vihma, 2018). More importantly, multi-hole probes are able to resolve all three wind velocity components. The simplest multi-hole probe capable of resolving all three velocity components is the five-hole probe, composed of five holes arranged symmetrically on a semi-spherical or conical tip. When the wind velocity is
oriented in different directions relative to the probe axis, each hole converts a different proportion of the dynamic pressure to stagnation pressure, allowing the dynamic pressure and direction to be determined using laboratory or in-flight calibration of the probe's directional response.

The use of five-hole probes in sUAS measurements has evolved from their employment in manned aircraft measurements (Lenschow, 1970, 1972). Such measurements frequently employ in-flight calibration procedures. For instance, Parameswaran
and Jategaonkar (2004) presented the calibration procedure of five-hole probes using flight recorder data using an optimization algorithm to estimate the time delay, biases and scale factors in the pressure measurements. Afterwards, the corrected five-hole probe measurements were compared with the measurements from the inertial measurement unit to check their compatibility. Drüe and Heinemann (2013) present a comprehensive review of in-flight calibration of several atmospheric measurement instruments, including five-hole probes. They identified five-hole probe in-flight calibration maneuvers to determine the sideslip
angle, angle of attack, static pressure and position errors. Moreover, they highlighted the need for in-flight calibration in the experiment location under favorable atmospheric conditions and following removal of the sensors from the aircraft.

The simplicity and compact nature of multi-hole probes also make it particularly useful for fixed-wing sUAS. However, as with manned aircraft, these aircraft necessarily fly at velocities an order of magnitude greater than the wind velocity, their





usage can be very sensitive to small errors in calibration and probe alignment (Suomi and Vihma, 2018; Laurence and Argrow,
2018). Furthermore, accurate position and orientation determination usually requires very accurate orientation information
(e.g. obtained through the use of dual-antenna combination GPS/IMU units) and accurate time-stamping of the data is critical
to align sensor and and flight data. Here, we present an approach which can be implemented *a posteriori* to minimize the
impact of unidentified and unquantified biases introduced during wind velocity measurement which result in contamination of
the wind signal by the aircraft velocity. In the following sections we overview a multi-probe implementation and discuss the
potential sources of bias within the approach. We then present a simple automated optimization which is designed to identify
and remove these biases and demonstrate that this approach improves the wind estimate of an existing dataset.

## 2   Multi-hole Probe Implementation

Multi-hole probes are an adaptation of the common Pitot-static probe to allow the determination of relative wind direction as
well as magnitude. Widely used in laboratory wind-tunnel studies of three-dimensional flow fields, they found use in manned-
aircraft studies of atmospheric wind (Treaster and Yocum, 1978; Axford, 1968; Lenschow, 1972) before being adopted for
sUAS use. Five-hole probes, being the simplest form of multi-hole probe, are most common. The arrangement of the normal
vector of each plane of the holes on the probe tip typically consists of a central hole with normal vector parallel to the probe
axis, measuring pressure $P_1$, and with the normal vector of the remaining holes at an angle (typically $20°$ to $45°$) to the probe
axis. Two holes measure the pressure at opposing directions on the horizontal plane e.g. $P_2$ and $P_3$, with the remaining two on
opposite directions on the vertical plane, e.g. $P_4$ and $P_5$. Static pressure, $P_s$ is also measured, either through a ring of holes
oriented perpendicular to the probe axis, or through an alternate pressure port. Wind tunnel calibrations are used to determine
calibration coefficients, for example

$$C_{P_{yaw}} = (P_2 - P_3)/(P_1 - \overline{P}) \tag{1}$$
$$C_{P_{pitch}} = (P_2 - P_5)/(P_1 - \overline{P}) \tag{2}$$
$$C_{P_{total}} = (P_1 - P_0)/(P_1 - \overline{P}) \tag{3}$$
$$\overline{P} = (P_2 + P_3 + P_4 + P_5)/4, \tag{4}$$

where $P_0 = 0.5\rho|\boldsymbol{U}_m|^2 + P_s$ is the total pressure, $\rho$ the density of the air and $|\boldsymbol{U}_m|$ the magnitude of the relative air velocity
vector $\boldsymbol{U}_m$. During calibration $P_1$, $P_2$, $P_3$, $P_4$, $P_5$ and $P_s$ are measured at different yaw, $\beta$, and pitch, $\alpha$, angles at known $P_0$.
Depending on the specifics of the probe geometry, a unique set of coefficients is recovered for each $\alpha$ and $\beta$ combination up
to some limit (referred to as the cone angle) typically between $25°$ and $45°$ depending on the specifics of the probe geometry.
During measurement $P_1$, $P_2$, $P_3$, $P_4$, $P_5$ and $P_s$ are simultaneously sampled and $C_{P_{yaw}}, C_{P_{pitch}}$ calculated for each sample. The
known dependence of $\alpha$, $\beta$ and $C_{P_{total}}$ on $C_{P_{yaw}}, C_{P_{pitch}}$ from the calibration is then applied to determine $\alpha$, $\beta$ and $P_0$ which,
when combined with a known $\rho$, provides the air velocity and direction relative to the probe axis. Additional calibration is also
possible to account for imprecise frequency response of the probes caused by resonance and viscous damping in the pressure





tubing and sensors (e.g. as described in Gerstoft and Hansen (1987)) which can potentially require additional corrections (e.g. Yang et al., 2006). These additional corrections are not addressed here.

When implemented on a moving platform such as an aircraft, $U_m$ is no longer the wind velocity but is instead a combination of the aircraft and wind velocity vectors

$$[U_m]_B = [U]_B - [U_s]_B \tag{5}$$

where $U_s$ is the velocity vector of the sensor and $U$ is the desired wind velocity vector. We have also introduced the subscript $B$ to indicate that these velocities are in a body-fixed frame of reference, i.e. a coordinate system attached to the aircraft. Due to the pitch, roll and yaw angles of the aircraft, $\Omega = [\theta\ \phi\ \gamma]$, or more specifically their time rate of change $\dot{\Omega}$, $U_s$ can experience additional velocity relative to the aircraft velocity $U_{ac} = [U_{ac}\ V_{ac}\ W_{ac}]$ such that

$$[U_s]_B = [U_{ac}]_B + [\dot{\Omega} \times r]_B \tag{6}$$

where $r$ is the distance vector between the aircraft center of gravity and the measurement volume on the probe.

Note that the desired quantity is $[U]_I = [U\ V\ W]$, the wind velocity vector in the Earth-fixed inertial frame of reference. Furthermore, $U_{ac}$ is also typically measured in the Earth-fixed inertial frame (e.g. through global positioning system) and therefore a transformation matrix $L_{BI}$ must be determined using the aircraft's pitch ($\theta$), roll ($\phi$), and yaw ($\gamma$) angles in the inertial frame. The velocity vector $[U_{ac}]_I$ along with $\theta$, $\phi$, $\gamma$ and their rates as required to build $\Omega$ and $L_{BI}$ can be measured

by an onboard inertial measurement unit and GPS, and are commonly provided by most autopilots used for sUAS operation. Thus, providing enough information to determine $[U]_I$ from $[U_m]_B$.

However, as noted earlier, $U_s$ is often an order of magnitude larger than the desired $U$, making the process sensitive to an abundance of small biases. For example, the procedure above assumes perfect alignment between the probe's coordinate system and the aircraft's coordinate system. It also assumes that the $\Omega$ and $L_{BI}$ are measured at the aircraft's center of gravity;

that $r$ is precisely known; that there are negligible flow blockage effects in the wind tunnel calibration or from the aircraft fuselage; and that all sensors are free of error. Although every effort can be made to minimize these biases, it is unlikely that they can be removed completely. The result is that $U$ often contaminated by $U_s$. This is most evident when $L_{BI}$, $U_{ac}$ and $\Omega$ are changing rapidly. The following section describes a procedure developed to determine additional calibration coefficients and implement them following an in-flight calibration or measurement to minimize these biases.

## 115   3   Correction procedure

The net effect of the majority of biases can be summarized as influencing four parameters. Misalignment of probe and aircraft axes, calibration errors, and aerodynamic distortion of the flow around the probe will introduce bias errors into the time-dependent pitch, roll and yaw angles $\theta(t)$, $\phi(t)$ and $\gamma(t)$, which relate the measured velocity vector in body-frame coordinates to the aircraft velocity vector. In addition, calibration errors, transducer errors, and airframe aerodynamic effects (e.g. airframe

blockage and streamline deflection due to the generation of lift ) can also influence the direction of airflow relative to the probe





as well as the magnitude of the measured dynamic pressure $Q(t) = 0.5\rho(t)|\boldsymbol{U}_m(t)|^2$ relative to the true dynamic pressure. Note that the distortion of the flow may also depend on lift production of the aircraft and therefore $Q(t)$ may also include dependence on lift coefficient, which is not considered in the version of the corrections described here. Finally, it is also important for all sensor readings to precisely correspond to orientation readings in time to allow precise removal of aircraft

motion from measured relative air velocity. However, implementation of software and differences in sensor time response can cause a delay between when probe measured velocity and aircraft measured velocity occur, e.g. the values of $\boldsymbol{U}_m(t)$ and $\boldsymbol{U}_s(t)$ may not correspond to the same $t$. The proposed correction procedure assumes that these values are biased in a way such that

$$\theta(t) = \theta_m(t) + \Delta\theta \tag{7}$$

$$\phi(t) = \phi_m(t) + \Delta\phi \tag{8}$$

$$\gamma(t) = \gamma_m(t) + \Delta\gamma \tag{9}$$

$$Q(t) = \Delta Q\, Q_m(t) \tag{10}$$

$$\boldsymbol{U}_m(t) = \boldsymbol{U}_m(t_m + \Delta t) \tag{11}$$

where the subscripted $m$ indicates the measured value. The objective is then to find $\Delta\epsilon = \{\Delta\theta,\, \Delta\phi,\, \Delta\gamma,\, \Delta Q,\, \Delta t\}$. Using assumptions about how these biases will impact $\boldsymbol{U}(t)$ allows determination of optimal values for $\Delta\epsilon$ which minimize this

negative behavior. With $\Delta\epsilon$ known, we are able to remove its influence on the final time-series of $\boldsymbol{U}(t)$.

The assumptions used here are relatively straightforward. The first being that any biasing of $\boldsymbol{U}(t)$ by $\boldsymbol{U}_s$ will result in $\boldsymbol{U}(t)$ having dependence on the direction of travel of the aircraft. The second assumption we make is that the vertical component of $\boldsymbol{U}(t)$ will be approximately zero in the mean. This assumption may not hold in flight over sloped terrain, in which case an alternative assumption may be needed.

The correction procedure, as implemented, follows a multistage approach used to optimize $\Delta\epsilon$. This multistage approach was implemented due to allow different objective functions to be used for different components of $\Delta\epsilon$. However, in practice, it is likely that a well-implemented single-stage optimization will achieve the same results.

The first step is to identify a portion of the flight which will be used to determine $\Delta\epsilon$. This portion should not include any significant acceleration or deceleration (e.g. takeoff or landing) and should include multiple changes of direction of the aircraft.

In addition, the portion should be long enough to ensure that unsteadiness in the mean winds, e.g. as introduced by thermals, are averaged out and that . Ideally, devoting a portion of the flight after takeoff to conduct calibration orbits for later use in this process would be desired. With this portion of flight identified, the determination of $\Delta\epsilon$ is found by an optimization process seeking to minimize an objective function, $\delta$, through iterative calculation of $[\boldsymbol{U}]_I$ (as described in Section 2).

Through perturbing $\Delta\epsilon$ and examining its influence on $\boldsymbol{U}(t)$ it was found that the standard deviation of the horizontal

components of $[\boldsymbol{U}(t)]_I$, specifically $U(t)$ and $V(t)$, were most sensitive to $\Delta t$ (due to the aircraft flight being predominantly in the horizontal plane) with values of $\Delta t$ as low as tenths or hundredths of a second contributing to large biases of the horizontal components of $[\boldsymbol{U}(t)]_I$. We thus first use a Nelder-Mead multidimensional unconstrained nonlinear minimization approach, implemented using the Matlab *fminsearch* command, to identify the value of $\Delta t$ which minimizes the objective function $\delta$,





defined as

$$\delta_U = \langle U \rangle|_{U_{ac}>0} - \langle U \rangle|_{U_{ac}<0} \tag{12}$$

$$\delta_V = \langle V \rangle|_{U_{ac}>0} - \langle V \rangle|_{U_{ac}<0} \tag{13}$$

$$\delta = \delta_U^2 + \delta_V^2. \tag{14}$$

Note that $\langle \, \rangle|_{U_{ac}>0}$ indicates an average conditioned on when the aircraft is flying with positive inertial velocity component $U_{ac}$. Likewise $\langle \, \rangle|_{U_{ac}<0}$ indicates an average of all values obtained when the aircraft inertial $U_{ac}$ velocity component is negative. The selection of $U_{ac}$ vs $V_{ac}$ for conditioning is arbitrary, and should be selected based on the actual flight trajectory flown. For flight trajectories without many trajectory changes, it was also found that the objective function

$$\delta = \langle (U - \langle U \rangle)^2 \rangle + \langle (V - \langle V \rangle)^2 \rangle \tag{15}$$

was equally effective, but relies on the assumption that the biases will act only to increase the fluctuations of the velocity signal at the probe. Here $\langle \, \rangle$ indicates an average over the entire portion of the flight used to find $\Delta\epsilon$. The resulting value of $\Delta t$ which minimizes $\delta$ is then implemented in the remaining optimization stages.

The second stage follows a similar approach. Noting that the mean value of the vertical component of $[\boldsymbol{U}(t)]_I$, i.e. $W(t)$, is most sensitive to $\Delta\theta$, we then find the value of $\Delta\theta$ that minimizes

$$\delta = \langle W \rangle \tag{16}$$

using $\boldsymbol{U}_m(t) = \boldsymbol{U}_m(t_m + \Delta t)$ as found above.

The remaining elements of $\Delta\epsilon$, specifically $\Delta Q$, $\Delta\phi$ and $\Delta\gamma$, are then found by minimizing $\delta$ as defined in Equation 14 using $\boldsymbol{U}_m(t) = \boldsymbol{U}_m(t_m + \Delta t)$ and $\theta = \theta_m + \Delta\theta$ as found in the preceding two stages.

Finally, to ensure that the values of $\Delta\epsilon$ determined using the latter optimization stages do not influence the values found during the earlier stages, $\Delta\epsilon$ is further refined by repeating the above three stages once again. In practice, this last step only influenced $\Delta\epsilon$ by one percent or less and likely can be omitted without loss of confidence in the final values of $\Delta\epsilon$.

# 4   Results

With $\Delta\epsilon$ known, the biases described by $\Delta\epsilon$ can be removed following Equations 7 to 11 prior to a final determination of $[\boldsymbol{U}(t)]_I$. To validate this correction procedure, we applied it to measurement data acquired during the LAPSE-RATE campaign described in de Boer et al. (2018). The data was acquired using two different five-hole-probe-equipped fixed-wing aircraft, with the aircraft, probe, and data reduction procedures described in detail in Witte et al. (2017). We first demonstrate the correction procedure in flights compared to a ground reference, followed by a demonstration of the improvements made to vertical profiles of the wind velocity and direction.



## 4.1 Comparison to ground reference

A key part of the LAPSE-RATE campaign was an intercomparison study between numerous sUAS measuring pressure, temperature, humidity, wind speed and wind direction. As detailed in Barbieri et al. (2019), the intercomparison was conducted

by flying the sUAS near the Mobile UAS Research Collaboratory (MURC) vehicle, which was equipped with a 15 m mast supporting reference instruments, including a sonic anemometer. For the fixed-wing aircraft used here, this comparison was performed by having the aircraft orbit the mast at 20 m above ground level with an orbit radius of 80 m. This orbit was performed for approximately 5 minutes before the aircraft ascended to 120 m to perform similar orbits for 2 minutes, then descending back to 20 m to resume the orbits around the tower for another 2 minutes before starting its landing pattern.

This circular flight pattern introduced a periodic variation in $\theta$, $\gamma$ and $\phi$, with the period corresponding to the time to complete an orbit (approximately 25 s). Although convenient for the measurement of atmospheric parameters at a single geographic location, these types of orbits consist of the worst-case scenario for the contamination of the measured wind direction by the biases discussed in Section 2.

The periodicity is clearly evident in the estimated horizontal wind velocity magnitude, $(U^2 + V^2)^{1/2}$, and direction, $\psi$, prior

to implementing the corrections, as shown in Fig. 1a and 1b respectively. Although the general trends of the measured wind velocity and direction time series follow that of the reference velocity and direction, the magnitude of the fluctuations are clearly contaminated by the aircraft velocity and direction. The period in the wind signal is consistent with the time required to orbit the fixed mast at 25 s. Note that in Fig. 1 only the two portions of the flight where the sUAS is at the same altitude as the reference sensors are presented.

The same time series are shown in Figs. 2a and 2b corrected following the procedure described in Section 3. The ten minutes of flight between 12:49 MDT and 12:58 MDT were selected to conduct the optimization of $\Delta\epsilon$. The the result of optimization was $\Delta\epsilon = \{\Delta\theta = -6.4°, \Delta\phi = 0.9°, \Delta\gamma = 2.1°, \Delta Q = 1.07, \Delta t = -0.045s\}$ which highlights the sensitivity of the estimated wind velocity and direction to even small deviations from ideal orientations. As shown in Fig. 2. The corrected signals are now largely free of the 25 s periodicity, although there is some evidence of contamination between 12:56 MDT and

12:58 MDT. When the aircraft returns to 18 m altitude for the second set of orbits (which were not included in the optimization) there is little evidence of aircraft velocity contamination in the wind estimates.

To provide a more quantitative comparison between the sUAS and reference measurement, we directly compare the velocity magnitude and direction measured at each instant a sample was made by the ground reference. This comparison is presented in Fig. 3 in which a perfect comparison would result in the straight line as indicated on these figures. Note that, a perfect

correlation should not be expected as the sUAS and reference sensor were not collocated. Also shown as a dashed line on these figures are the bounds described by 2 standard deviations of the difference between the sUAS and MURC measured values.

For the uncorrected velocity magnitude and direction, the comparison shown in Figs. 3a and 3b reveals a broad spread about the reference line. This spread decreases significantly when the corrections are applied, as shown by comparison to Figs. 3c and 3d. The mean difference between the two measurements decreases by approximately 35% in magnitude and direction

with correction, corresponding to an increase in the correlation coefficient from 0.13 to 0.19 in magnitude and 0.22 to 0.32





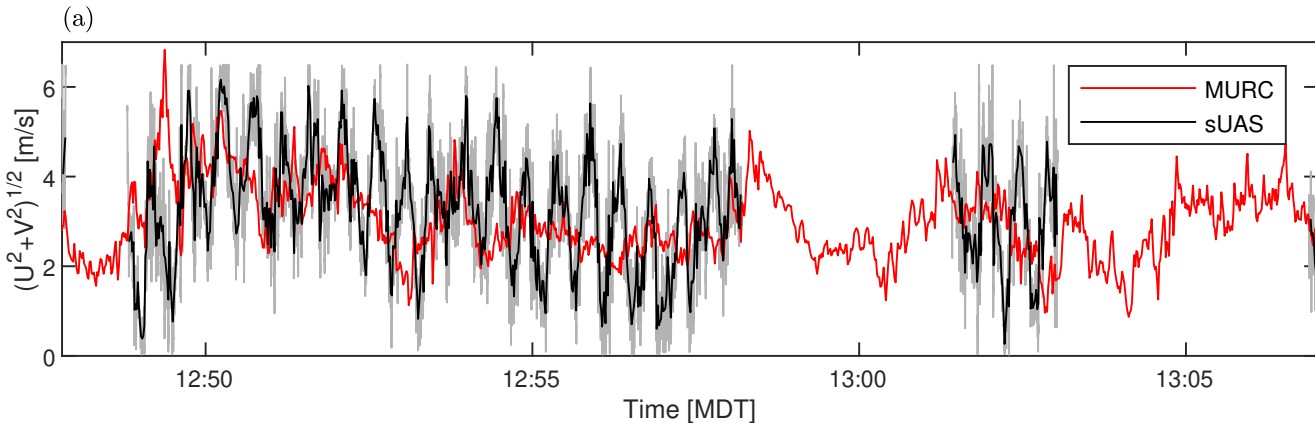

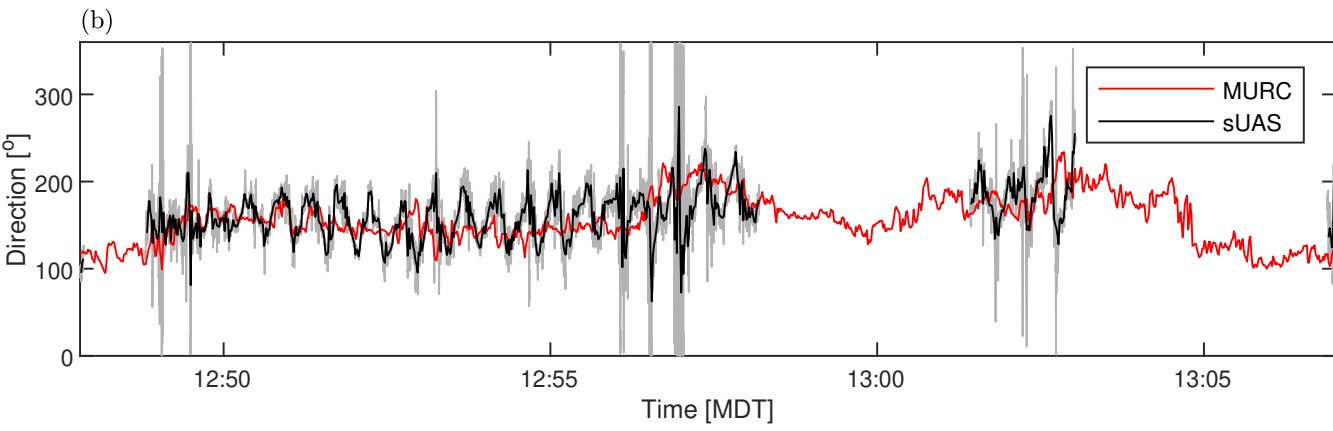

**Figure 1.** Comparison between uncorrected (a) horizontal wind velocity magnitude and (b) wind direction measured by sUAS to the reference signal measured by MURC. Gray lines indicate full signal from sUAS, whereas black lines indicate same signal downsampled to same data rate as that of the MURC using a simple moving average approach.

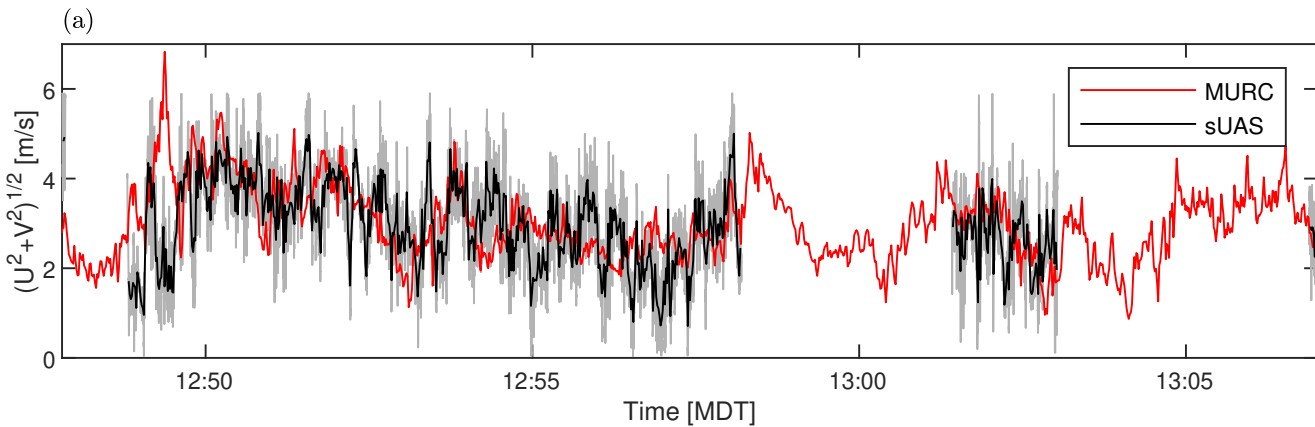

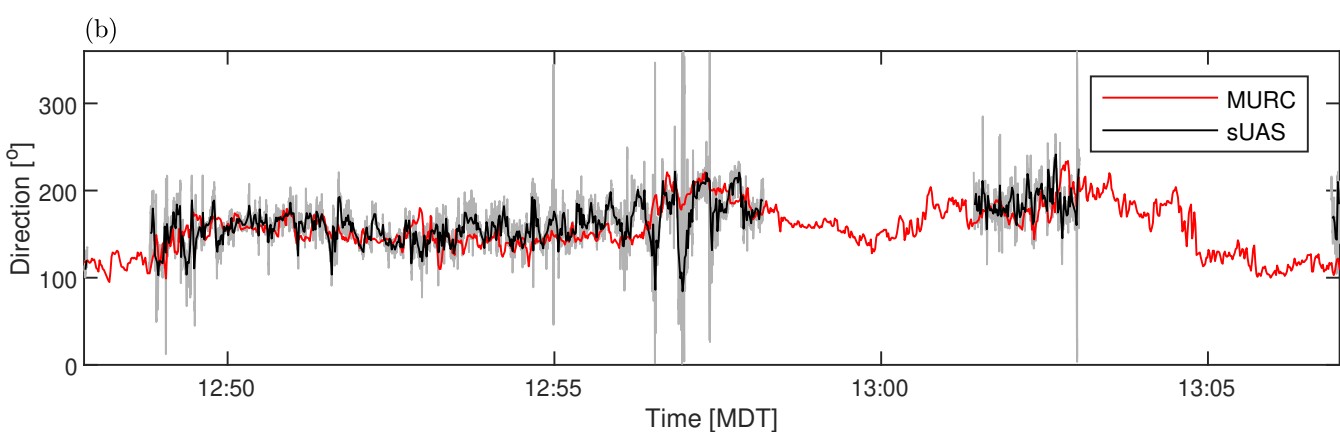

**Figure 2.** Small unmanned aerial system-based wind measurements from Fig. 1 following correction compared to the reference signal measured by MURC: (a) horizontal wind velocity magnitude and (b) wind direction. Gray lines indicate full signal from sUAS, whereas black lines indicate same signal downsampled to same data rate as that of the MURC using a simple moving average approach..

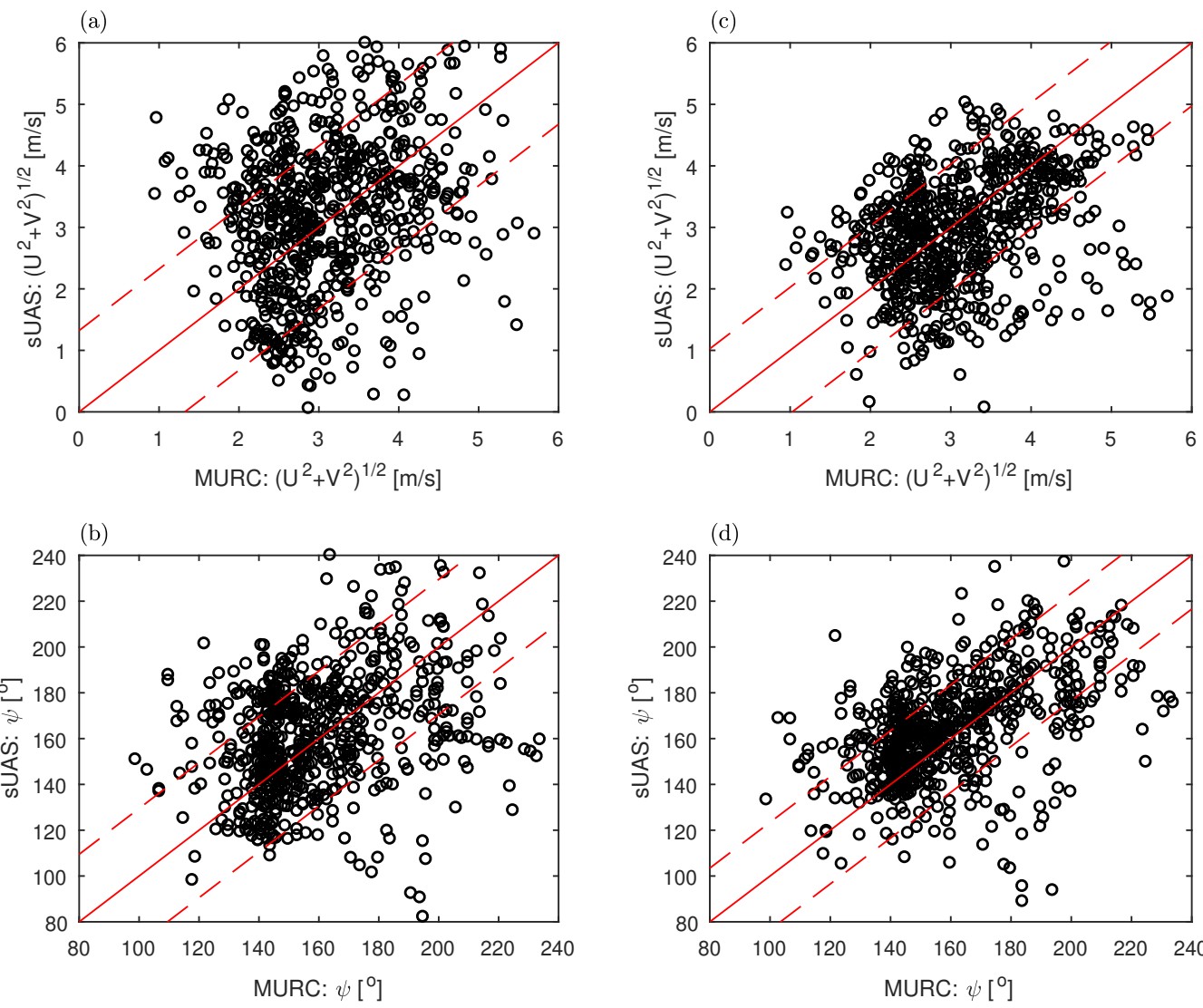

**Figure 3.** Direct comparison between (a) uncorrected and (b) corrected horizontal wind magnitude measured simultaneously by sUAS and MURC. Similar comparison shown for (c) uncorrected and (d) corrected wind direction. Solid red line indicates line where both measurements identical and dashed lines indicate 2 standard deviations of the difference between the sUAS and MURC measured values.





in direction. This increased correlation is reflected in the statistics. The correction brings the standard deviation of the sUAS measured velocity much closer to the reference signal. The standard deviation in magnitude measured by the sUAS decreases from 1.2 m/s to 0.90 m/s, very close to the value of 0.86 m/s measured by the reference. For direction the standard deviation decreases with correction from $27°$ to $22°$ whereas it is $24°$ for the reference.

## 4.2 Implementation in profiling measurements

The results of the comparison to the ground reference provide confidence in the success of the correction. To demonstrate the improvement offered by application of these corrections on vertical profiling by fixed-wing aircraft, we now examine their impact on profiles of wind speed and direction measured by two separate aircraft at different locations. These two fixed-wing aircraft were essentially identical in configuration to that described in Witte et al. (2017) and were flown at measurement sites separated by 16 km. Each aircraft measured an atmospheric profile every hour, with the two aircraft staggered in time by 30 minutes.

Each profile consisted of a 20 minute flight, with the aircraft performing a spiralling ascent to 900 m followed by a spiralling descent, with this pattern repeated until the 10Ah battery was expended. In the following discussion the times are those corresponding to when the profile measuring flight initiates with the $X, Y, Z$ coordinate system's origin at the takeoff location. These particular profiles were selected for discussion as they were measured during the boundary layer transition, and represent different behaviors, including the presence of turbulence and variability in the wind direction. The wind speeds during these profiles were also low, producing a large ratio of aircraft velocity to wind velocity and therefore a challenging case to accurately extract the wind components from the five-hole-probe signal.

As previously mentioned, the orbital flight patterns also represented a challenge for extracting the wind data due to the periodic variation in $\theta$, $\gamma$ and $\phi$ introducing a corresponding periodic variation in $[\boldsymbol{U}(t)]_I$. This bias can be clearly illustrated by comparing $\gamma$ to $\psi$, as done for an example flight in Fig. 4a. For this flight, the aircraft completed a full $360°$ turn in approximately 30 s, introducing a corresponding period in both the wind speed and direction before correction. Once the corrections have been applied, as shown in Fig. 4b, There is little-to-no evidence of this periodicity in the direction of the wind measured by the sUAS.

These corresponding wind speed and direction profiles are presented in Fig. 5 for sUAS 1 and Fig. 6 for sUAS 2. In these figures both the uncorrected and corrected profiles are displayed in order to show the relative improvement offered by application of the bias correction. For all profiling flights, the correction coefficients were determined by optimizing using the entire flight once the aircraft was in its flight pattern. Before correction, the bias introduced by the aircraft trajectory is apparent as large coherent deviations from the general trend, mostly evident in the velocity magnitude, but also present in the direction. When the corrections were applied, these large deviations were greatly reduced, better representing the structure of the boundary layer throughout the profiles. In the wind velocity profiles presented in Fig. 5 for sUAS 1 there were still high velocity deviations even in the corrected profiles near the surface corresponding to when the aircraft was being manually controlled and experiencing strong accelerations. It has been found that the corrections presented here cannot completely remove the bias due to aircraft acceleration, suggesting a potential time response lag between the five-hole probe and inertial measurement unit.



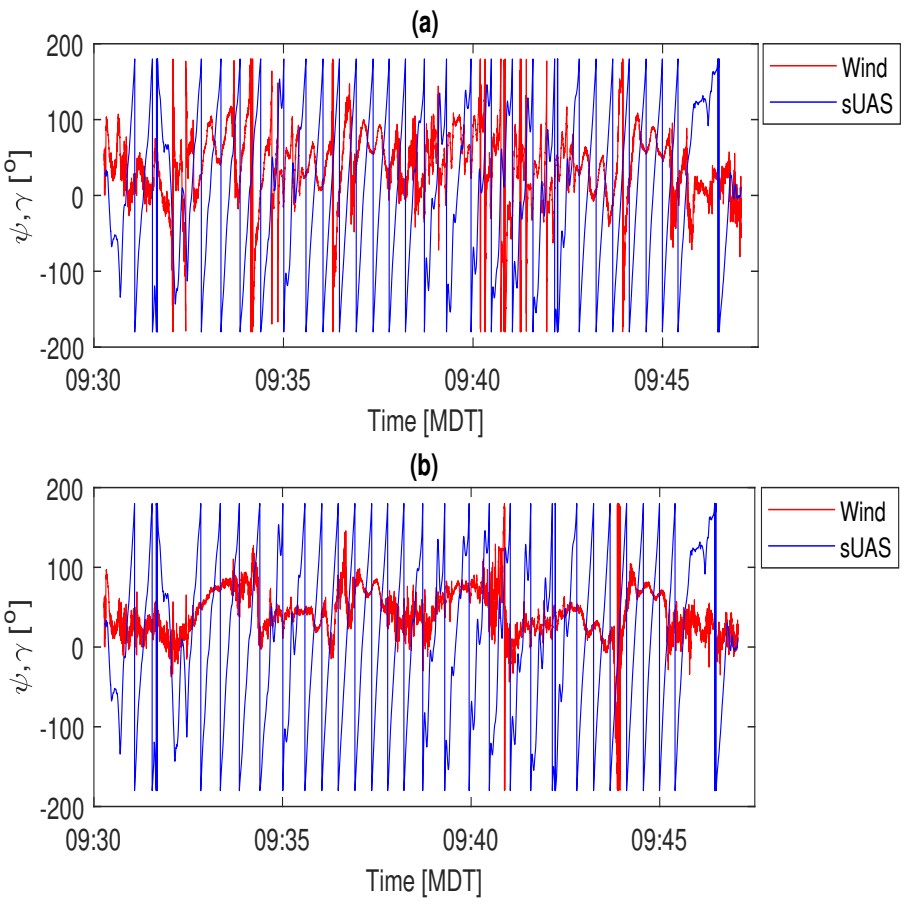

**Figure 4.** Comparison between measured wind direction and aircraft yaw direction for (a) uncorrected and (b) corrected signals as a function of time for a single flight.

The corrected profiles show very different wind behaviors existed for the different sites and times. At the site measured by sUAS 1, the profiles measured at 8:30 MDT and shown in Figs. 5a,b reflect the correspondence between the stable thermodynamic conditions throughout the boundary layer and the horizontal wind magnitude, with winds increasing from 2 m/s near the ground to 4 m/s at $Z = 900$ m and consistently between $\psi = 0°$ and $100°$. There was noticeably stronger velocity and direction fluctuations measured for $Z < 200$ m, indicating the presence of turbulence near the surface. This turbulence appears to have

been still present at 9:30 MDT, as shown in Figs. 5c,d, but at this time there was a region of calm air centered at $Z = 600$ m, coinciding to a significant deviation in measured wind direction. For $Z > 200$ m the corrected profile of wind direction was consistent with the one measured at 8:30 MDT, excepting the region of calm air at $Z = 600$ m.

At the site measured by sUAS 2, the corrected wind speed and direction profiles measured at 9:10 MDT and shown in Figs. 6a,b reflect a boundary layer undergoing transition, with evidence of turbulence for $Z < 500$ m. At 10:10 MDT, as shown

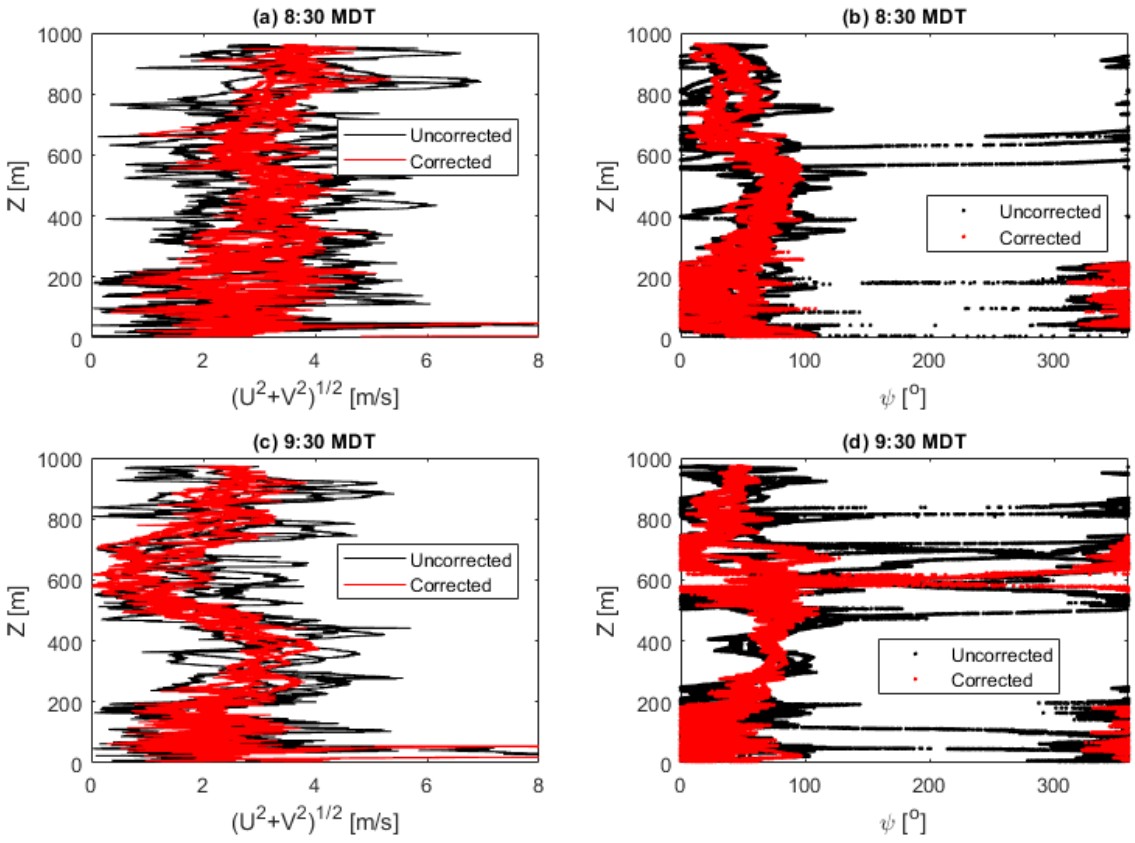

**Figure 5.** Comparison of (a) horizontal wind magnitude and (b) wind direction profiles measured by sUAS 1 at 8:30 MDT with and without correction applied. Horizontal magnitude and direction profiles measured at 9:30 MDT shown in (c) and (d) respectively.

in Figs. 6c,d, a multi-layer structure was also evident in the wind profiles in the form of significant changes in the wind direction throughout the profile. The horizontal wind velocity magnitude was relatively consistent between 1 to 2 m/s for $Z < 800$ m, but there was evidence of stronger turbulence for $Z < 500$ m and moderate wind shear for $Z > 700$ m. As noted, the wind direction exhibited significant variation in the range 400 m$< Z < 500$ m, with continual backing within 500 m$< Z < 900$ m, and veering for $Z < 900$ m. These different altitudes of behavior were consistent with the measured potential temperature

changes (not included for conciseness), and became much easier to identify in the corrected profiles than they were before correction.

It is clear through comparison of the corrected and uncorrected profiles in Figs. 5 and 6 that the corrections work under different conditions and for different aircraft. Similar improvements have been observed for other measured profiles measured with these and other sUAS. For the coefficients determined by the optimization routine for these profiles are presented in

Table 1. Comparing each flight by the same sUAS demonstrates that the automated optimization converged on nearly identical



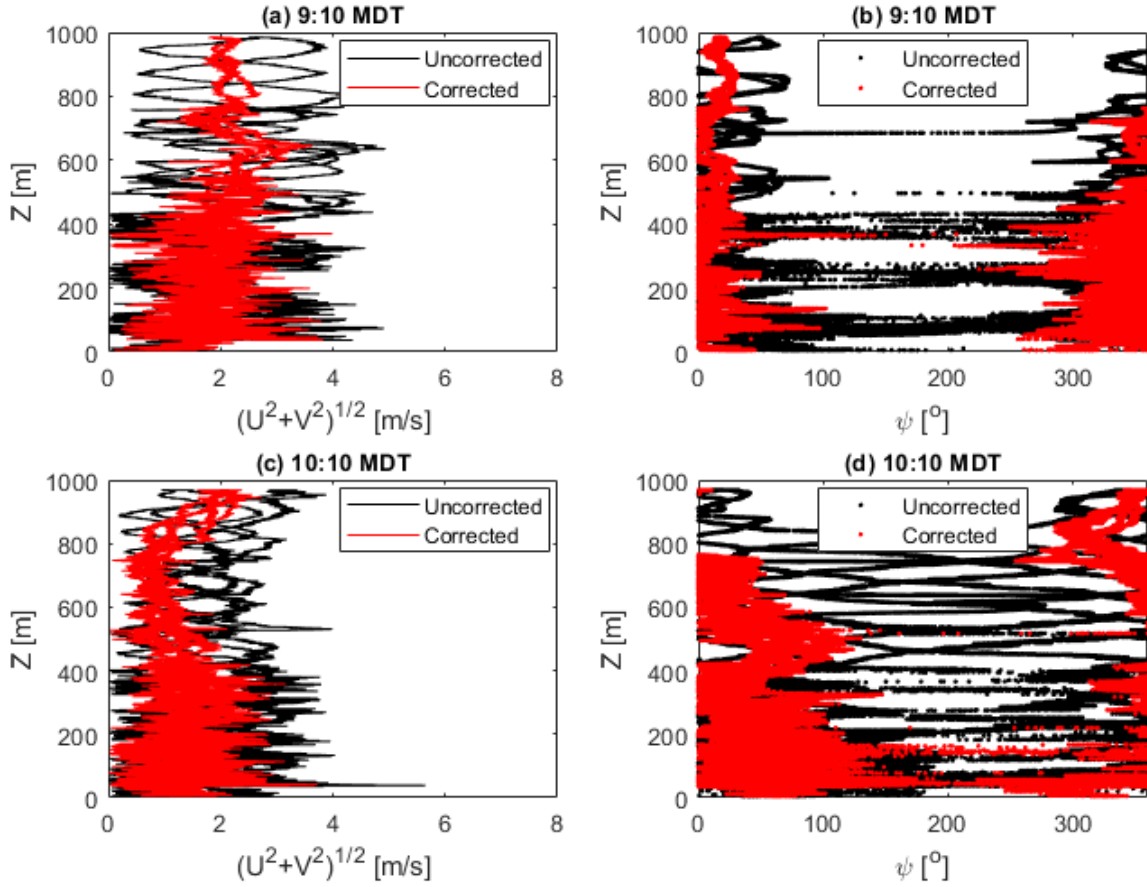

**Figure 6.** Comparison of (a) horizontal wind magnitude and (b) wind direction profiles measured by sUAS 2 at 9:10 MDT with and without correction applied. Horizontal magnitude and direction profiles measured at 10:10 MDT shown in (c) and (d) respectively.

| sUAS | Flight | $\Delta\theta$ | $\Delta\phi$ | $\Delta\gamma$ | $\Delta Q$ | $\Delta t$ |
|------|--------|--------|--------|--------|--------|--------|
| 1 | 8:30 MDT | -4.9° | 0.17° | 2.6° | 1.1 | 0.98 s |
| 1 | 9:30 MDT | -5.8° | 2.7° | 2.1° | 1.1 | 0.01 s |
| 2 | 9:10 MDT | -3.9° | 0.85° | 1.4° | 1.15 | 2.9 s |
| 2 | 10:10 MDT | -3.9° | 0.78° | 1.3° | 1.15 | 2.6 s |

**Table 1.** Components of $\Delta\epsilon$ determined by optimization for each sUAS for both flights.





coefficients for the same sUAS with only one coefficient changing by more than $1°$ between each flight. Indeed, the correction coefficients were found to be remarkably similar for each sUAS used throughout the LAPSE-RATE campaign. This similarity between the coefficients reinforces the assumption that the biases are caused by physical misalignment between the coordinate systems of the aircraft and five-hole probe. Note that bias corrected by $\Delta t$ should not be expected to be consistent for the systems used here, as the time series of $\boldsymbol{U}_m$, $\boldsymbol{U}_{ac}$ and $\boldsymbol{\Omega}$ are measured by separate acquisition systems at different rates and aligned using post-processing software. Thus the $\Delta t$ bias is most likely introduced by errors in this alignment process and can be expected to be random.

## 5 Conclusions

Small unmanned aerial systems have increasingly used in atmospheric research. Frequently, this research requires the acqui-
sition of the wind velocity vector. Multi-hole probe measurements are among the more common and reliable techniques used for this purpose. However, when implemented on sUAS there is significant potential to introduce bias due to the large ratio of aircraft velocity to the wind velocity. Therefore, the measured wind velocities are very sensitive to these small biases. Fur-
thermore, when conducting vertical profiles at a fixed location, these profiles typically require circular flight patterns which increase the probability of small misalignment between the probe and the aircraft axes introducing a time-dependent, periodic error in the wind velocity measurement that can propagate into post-flight analysis such as the calculation of energy spectra and Reynolds stresses.

An approach was presented that can be applied in post-processing of the flight data to automatically estimate the biases in axis misalignment, as well as errors in their alignment in time. Once estimated, these biases can be removed, improving the quality of the wind estimate.

These corrections were validated using data acquired as part of the LAPSE-RATE field campaign. Measurements flown near a ground-based reference system revealed significant reduction in measured oscillations of both wind magnitude and direction, which corresponded to the aircraft flight pattern. Additional verification was conducted by comparing profiles of wind speed and direction measured by two different aircraft at two different times. The estimated biases were within $\pm 1°$ for each aircraft, and successful minimization of aircraft-induced oscillations in the measured profiles was observed for both aircraft. These results confirm that the biases are most likely due to physical misalignment of the aircraft and probe axes, as well as demonstrating that the same correction procedures can be applied to multiple aircraft.

*Author contributions.* S.B. conceived the correction, which was implemented by L.A-G. Both authors contributed to manuscript preparation.

*Competing interests.* The authors declare no competing interests are present.



*Acknowledgements.* This work was supported by the National Science Foundation through grant #CBET-1351411 and by the National
Science Foundation Award No.1539070, Collaboration Leading Operational UAS Development for Meteorology and Atmospheric Physics
(CLOUDMAP). The authors would like to thank Caleb Canter, Jess Estridge, Jonathan Hamilton, Sean MacPhee, Ryan Nolin, Isaac Rowe,
Christopher Saunders, Virginia Smith, Christina Vezzi and Harrison Wight who maintained and flew the aircraft used in this study, as well as
calibrated and manufactured the probes.



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
