# Peer review of "An Approach to Minimize Aircraft Motion Bias in Multi-Hole Probe Wind Measurements made by Small Unmanned Aerial Systems"

_Atmospheric Measurement Techniques, 2020_

## Short Comment (SC1) · 30 Jun 2020

Hi, this is my first comment, I hope this is ok... Wouldn't a FFT analysis of the data presented in figure 1 and 2 nicely show the removal of periodic data? You could include groundspeed of the UAV, airspeed, and measured wind speed (the latter of both UAV and met mast). Then you could do the same again after enabling your correction algorithms. The common frequency peak in groundspeed and UAV measured windspeed should be removed.

[Figure]

All the best & success with your publication, William

---

## Author Comment (AC1) · 1 Jul 2020

Thank you for your comment. Although spectral analysis is one of the metrics we used to validate the approach during development, it is not an intrinsic part of the correction. We initially did not include spectral analysis due to the need to identify and clarify the influence of large-scale flow contributions in the spectrum, spectral roll-off from turbulence, and tubing response contributions.

However, in retrospect, we realize we probably should have included at least some

example spectral analysis. We have attached a figure to this comment that we will introduce into the paper if we are invited to submit a revision. The figure shows the power spectral density calculated from the same flight data shown in Figure 1 and 2. It shows that a spectral peak at (1/25=0.04 Hz: corresponding to the orbital period of the aircraft) is significantly reduced by the correction.

[Figure]

[Figure]

**Fig. 1.**

---

## Referee Comment (RC1) · Anonymous Referee #2 · 23 Jul 2020

general comments:

Reliable wind and turbulence measurements are highly required for a wide range of topics in basic atmospheric boundary layer research and its application, e.g. for air quality estimates or the wind field characterization for wind energy purposes. UAS provide a flexible and rapidly increasing resource for performing those measurements in unprecedented spatio-temporal resolution. The presented manuscript provides a step forward in the understanding of inherent errors in the wind estimates from multi-hole probes mounted on small aircraft, and also suggests and validates a method to mini-

mize those uncertainties. By that the manuscript is timely and thematically well-suited for being considered for publication in AMT. There are, however, a few shortcomings that have to be addressed before I can recommend publication. The manuscript is in general well written and clearly structured. One exception is the introduction that in my opinion misses a clear line of argumentation towards the main goals of the presented work. The figures are in general of rather good quality, except for 7 and 8 that are nearly impossible to read and interpret and need a thorough overhaul (see also my specific comments below). Based on that I recommend major revisions to the manuscript in its present form.

specific comments:

l2: "aircraft spatial orientation, translation and velocity"; I feel there is some inconsistency/inaccuracy that should be clarified: a) Aircraft spatial orientation: do you mean attitude with respect to the Eulerian angles for Pitch, roll and yaw or something else? b) translation is already a velocity, do you want to distinguish between translational and angular velocities? be more Clear and concise here!

l12: insert "a" before "wide"

l13: insert "," before "such as"

l15: you are citing a lot here, which is in general not bad, but if you decide to go so broad out, then I feel that in particular for turbulence there are some central references missing, e.g.:

Mansour, M., Kocer, G., Lenherr, C., Chokani, N., & Abhari, R. S. (2011). Seven-Sensor Fast-Response Probe for Full-Scale Wind Turbine Flowfield Measurements. Journal of Engineering for Gas Turbines and Power, 133(8), 081601. https://doi.org/10.1115/1.4002781

Calmer, R., Roberts, G. C., Preissler, J., Sanchez, K. J., Derrien, S., & O'Dowd, C. (2018). Vertical wind velocity measurements using a five-hole

probe with remotely piloted aircraft to study aerosol–cloud interactions. Atmospheric Measurement Techniques, 11(5), 2583–2599. https://doi.org/10.5194/amt-11-2583-2018

Båserud, L., Reuder, J., Jonassen, M. O., Kral, S. T., Paskyabi, M. B., & Lothon, M. (2016). Proof of concept for turbulence measurements with the RPAS SUMO during the BLLAST campaign. Atmospheric Measurement Techniques, 9(10), 4901–4913. https://doi.org/10.5194/amt-9-4901-2016 those references should also be used again when introducing multihole probes on sUAS, i.e. p2, l40/41

l22: what is the flexibility in a profile? do you mean the flexibility in chosing ascent/descent rates, or just the flexibility in location that is already stated before?

l26: I think you forgot temperature as the most commonly sampled parameter!

l37/38: "Usually, wind velocity measurements by fixed-wing sUAS require velocity probes with slightly better temporal response than Sonic anemometers (Witte et al., 2017; Mayer et al., 2012)."; My first response here was "Why?"; what exactly do you want to Express here; there are sonics around with 100 Hz measurement capability matching the typical sampling frequency of multihole probes

l42: "large wind velocities"; that should here better read "large flow velocities", as we are talking about the relative flow between the probe and the air

l49: "fly at velocities an order of magnitude greater than the wind velocity"; that can of course be the case, but in reality you easily can fly a fixed-wing in wind speeds up to 80% of the cruise speed of the Aircraft, so you should not generalize this statement

l60: replace "multi-hole probe" by "multi-hole probes"

l61: remove "with" before "a central hole"

l61: I suggest to replace "parallel to" by "in line with"

l85: Equation 6 and corresponding text: that reminds me very strongly to Don

Lenshows basic work; maybe a good idea to refer to!

l116: insert "the" before "determination"

l123: remove "due" before "to allow

l126: "... and should include multiple changes of direction of the aircraft."; How relates this statement to the calibration maneuvers suggested/required by Lenschow?

l175 (and other occasions): "horizontal wind velocity magnitude"; I strongly suggest to use "horizontal wind speed" instead, this is the meteorologically correct term here

figure1: has apparently the wrong y-axis label (should be wind direction in degrees); I also highly recommend not to use line plots for wind direction!; how is the downsampling done, just picking an individual value or applying some form of averaging? in addition I would just use horizontal wind speed, maybe abbreviated as vh as y-label for a)

figure2: same comments as for figure 1:

figure 3: again the velocity labels could be much easier and intuitive vh_MURC and vh_sUAS; in addition there is something mixed up in the figure caption: Wind speed is a) and c), not a) and b) as stated in the caption

l219: "there is little-to-no evidence of this periodicity! is a rather brave statement; I still see some clear indications of such a periodicity and a formulation in the direction "shows a distinctly reduced periodicity" sound to me much more appropriate!

figure6 and discussion l221-227; what exactly do you want to achieve with this potential temperature profiles? If it is just to give an overview on the state of the atmosphere, then you should distinctly simplify your presentation by e.g. only showing the 4 average profiles (e.g by bin averaging over 25 m vertical intervalls; if you are also interested to present an inter-flight variability you can achieve that by using whiskers around your bin mean); as it is it is a rather hard to read/interpret figure

figures 7 and 8: those are just messy in the present form; if you want to keep the shown information you could have this as a background with grey and reddish color, but on top you should again show some bin averaged values that would then give a clear picture how mean value and variability/standard deviation react on the proposed correction.

l235: insert "," before "even"

l237: "suggesting a potential time response lag between the five-hole probe and inertial measurement unit" a) insert "the" before "inertial measurement unit" b) this has been reported before: Båserud, L., Reuder, J., Jonassen, M. O., Kral, S. T., Paskyabi, M. B., & Lothon, M. (2016). Proof of concept for turbulence measurements with the RPAS SUMO during the BLLAST campaign. Atmospheric Measurement Techniques, 9(10), 4901–4913. https://doi.org/10.5194/amt-9-4901-2016

l244: insert "," after "For Z>200 m"

l256: remove "measured" before "profiles"

l264/265: how will a systematic time shift, e.g. introduced by a time delay of the data output of the IMU (as hypothesized in line 237), affect the correction procedure and your results? It might be worth to test (e.g. by a correlation analysis) if there is such a systematic time delay in your data set.

l267: insert "been" after "increasingly"
* * *

---

## Referee Comment (RC2) · Anonymous Referee #3 · 22 Sep 2020

Content:

The paper presents an introduction on where and why sUAS are used in atmospheric wind measurements, how MHPPs are calibrated and how they are used to determine the wind vector. The most important errors in this measurement are a misalignment of the probe and the aircraft axes, calibration errors, aerodynamic distortion by probe and aircraft body (which might also depend on the lift coefficient), transducer errors and time synchronization errors. A correction procedure for uncertainties in the roll, pitch and yaw angle alignment, as well as for the measurement of the dynamic pressure and

time synchronization are presented. The correction assumes that vertical wind speed is zero on average.

I am surprised that there is such a large cross-talk between ground speed or aircraft attitude and wind speed in fixed wing aircraft. However, I have experience with wind-measuring rotary wing devices only, where these problems seem to be much smaller, most likely due to smaller vehicle velocities. Suitable correction algorithms for fixed-wings seem to be particularly important. The paper therefore addresses relevant scientific questions and is suitable for publication in AMT. I wonder why there is no example dataset and example correction code available. In my opinion, this must be the case.

Specific comments:

Line 26: Temperature? (e.g. Witte2017)

Line 32: The difference between the three approaches is not clear: An onboard wind sensor measures air speed, and aircraft kinematics are used to determine ground speed. This seems to be the same as the second approach that you mention. Please briefly explain the differences, it might be helpful to add a reference for each approach (the first approach lacks a reference).

Line 34: Typo "kinmatic".

Line 36: Sensor-based wind measurements: Isn't everything that measures wind sensor-based? An IMU can be used to determine wind, but it is also a sensor (typically it consists of even 3x3 sensors). Which sensors do you mean?

Line 38: Witte2017 writes "Typically, these measurements employ wind velocity probes with a temporal response that is little better than that of sonic anemometers", and "Increasingly, UAVs are utilizing five-hole pressure probes [32,33,40], which can resolve to 40 Hz while flying at approximately 20 m/s.". Today's 3D sonic anemometers can have a data output rate of 100 Hz (e.g. Gill R3-100). So, I am not sure if this is true

anymore.

Multi-hole probe implementation: very clear

Line 142: A change in direction (what direction? Flight direction? Yaw angle?) will result in an acceleration due to a curvature of the flight path. So what kind of acceleration (rate of change of velocity) do you mean? Flight velocity changes? Vertical acceleration?

146: incomplete sentence

I can not scientifically judge the appropriateness of the optimization algorithm presented in section 3, however, it seems appropriate to me.

Line 194: When you argue with periodicity, then why not show it in a FFT plot?

Line 202: Does delta_Q have a unit?

Why are figures 5+6 bitmaps and not vector graphs? Is there a way to omit the wrapping-around at 360°? Is there a better way to convince the readers that the correction improves the accuracy of the data? Because the true velocities are apparently unknown, I would again prefer spectral analyses, that show that the motion of the aircraft becomes less apparent in the corrected data. Color schemes in figures might be better if same colors are used for same objects (e.g. red = sUAS and black = reference). Please also check that colors correctly convert to a gray scale that is distinguishable in black and white print outs.

Line 250-266: This seems to be a discussion of the specific weather conditions of that day on that site, I don't see how this adds to the message of the manuscript. Please explain.

Line 267: "Corrections work": do they improve the data? How do you prove this?

Line 279: is there a word missing in the first sentence?

---

## Author Comment (AC2) · 2 Oct 2020

We would like to thank the reviewer for their time spent in review of this manuscript and providing their detailed comments and suggestions. We have prepared a revised manuscript which we believe addresses the concerns and comments raised in their review. In the revised manuscript, specific changes are indicated in blue text, with each of the reviewer's comments addressed below.

1. 1. The introduction in my opinion misses a clear line of argumentation towards

[Figure]

the main goals of the presented work.

The authors added the following statement to strengthen the connection to our main goals: "This contamination results in over- or under- estimation of the wind vector and, in particular, errors in estimation of turbulence statistics (e.g. momentum fluxes, dissipation rate, turbulence kinetic energy) measured by the sUAS. Hence, it is vital to minimize errors in the wind components measured by sUAS."

2. l2: "aircraft spatial orientation, translation and velocity"; I feel there is some inconsistency/inaccuracy that should be clarified: a. Aircraft spatial orientation: do you mean attitude with respect to the Eulerian angles for Pitch, roll and yaw or something else?

Yes, we were referring to Euler angles (pitch, roll and yaw) in the present case. The statement as written was intended to be a more general statement as other approaches can also be used to describe the aircraft orientation in Earth-fixed coordinates. We have revised this sentence to explicitly mention Euler angles

b. translation is already a velocity; do you want to distinguish between translational and angular velocities? be more clear and concise here!

The authors have revised the sentence as: "Multi-hole probe mounted on an aircraft provide the air velocity vector relative to the aircraft, requiring knowledge of the aircraft spatial orientation (e.g. Eulerian angles), translational velocity, and angular velocity to translate this information to an Earth-based reference frame and determine the wind vector."

3. l12: insert "a" before "wide"

The authors have added it.

4. l13: insert "," before "such as"

The authors have added it.
5. l15: you are citing a lot here, which is in general not bad, but if you decide to go so broad out, then I feel that in particular for turbulence there are some central references missing, e.g.:

   - Mansour, M., Kocer, G., Lenherr, C., Chokani, N., & Abhari, R. S. (2011). Seven-Sensor Fast-Response Probe for Full-Scale Wind Turbine Flowfield Measurements. Journal of Engineering for Gas Turbines and Power, 133(8), 081601. https://doi.org/10.1115/1.4002781
   - Calmer, R., Roberts, G. C., Preissler, J., Sanchez, K. J., Derrien, S., & Oapos;Dowd, C. (2018). Vertical wind velocity measurements using a five-hole probe with remotely piloted aircraft to study aerosol–cloud interactions. Atmospheric Measurement Techniques, 11(5), 2583–2599. https://doi.org/10.5194/amt-11-2583-2018
   - Båserud, L., Reuder, J., Jonassen, M. O., Kral, S. T., Paskyabi, M. B., & Lothon, M. (2016). Proof of concept for turbulence measurements with the RPAS SUMO during the BLLAST campaign. Atmospheric Measurement Techniques, 9(10), 4901–4913. https://doi.org/10.5194/amt-9-4901-2016

   those references should also be used again when introducing multihole probes on sUAS, i.e. p2, l40/41

   The authors have added the references as suggested by the reviewer.

6. l22: what is the flexibility in a profile? do you mean the flexibility in choosing ascent/descent rates, or just the flexibility in location that is already stated before?

   What we meant by the profile flexibility is the ability of the sUAS to fly in different patterns easily compared with large manned aircraft. We have changed profile to flight path in the text.

7. l26: I think you forgot temperature as the most commonly sampled parameter!

   Definitely an oversight! The authors have added temperature to the list.

8. l37/38: "Usually, wind velocity measurements by fixed-wing sUAS require velocity probes with slightly better temporal response than Sonic anemometers (Witte et al., 2017; Mayer et al., 2012)."; My first response here was "Why?"; what exactly do you want to express here; there are sonics around with 100 Hz measurement capability matching the typical sampling frequency of multihole probes.

We were trying to highlight the utility of five-hole-probes relative to other available technology. Although such high frequency sonic anemometers do exist, to our knowledge, these anemometers do not exist in a package size sufficiently lightweight to be used on a fixed-wing sUAS. Hot-wire anemometry is also an option, but is not yet in common usage for sUAS measurements and, to our knowledge, multi-sensor hot-wire anemometry capable of resolving the velocity vector has not yet been utilized on sUAS. The authors have removed this sentence as the justification for using multi-hole probes was provided in the sentence following the one described above, and no further justification for their use relative to sonic anemometers is required.

9. "large wind velocities"; that should here better read "large flow velocities", as we are talking about the relative flow between the probe and the air

The authors have replaced "large wind velocities" by "large flow velocities" as the reviewer suggested.

10. l49: "fly at velocities an order of magnitude greater than the wind velocity"; that can of course be the case, but in reality you easily can fly a fixed-wing in wind speeds up to 80% of the cruise speed of the aircraft, so you should not generalize this statement

It's hard to imagine our aircraft successfully flying in the ABL turbulence produced by 16 m/s winds, but the reviewer is correct and we have revised the sentence as follows: "These aircraft have the ability to fly at velocities up to an order of magnitude greater than the wind velocity."

11. l60: replace "multi-hole probe" by "multi-hole probes"

The authors have replaced "multi-hole probe" by "multi-hole probes".

12. l61: remove "with" before "a central hole"

The authors have removed "with".

13. l61: I suggest to replace "parallel to" by "in line with"

The authors have replaced "parallel to" by "in line with".

14. l85: Equation 6 and corresponding text: that reminds me very strongly to Don Lenshows basic work; maybe a good idea to refer to!

The authors have added the reference to Lenschow's work.

15. l116: insert "the" before "determination"

The authors have added "the".

16. l126: "... and should include multiple changes of direction of the aircraft."; How relates this statement to the calibration maneuvers suggested/required by Lenschow?

Lenschow and several scholars suggested different calibration maneuvers to correct the sideslip angle and the angle of attack of multi-hole probes as reported by (Drue and Heinemann, 2013) to account for the flow distortion around the aircraft body. The maneuvers reported vary from straight flights to race tracks and other patterns depending on the parameters that needed to be corrected. These same maneuvers would be suitable for the correction described in this manuscript.

17.  (a) l175 (and other occasions): "horizontal wind velocity magnitude"; I strongly suggest to use "horizontal wind speed" instead, this is the meteorologically correct term here The authors have replaced the term "horizontal wind velocity magnitude" by "horizontal wind speed" throughout the text.

(b) figure1: has apparently the wrong y-axis label (should be wind direction in degrees); I also highly recommend not to use line plots for wind direction!; how is the downsampling done, just picking an individual value or applying some form of averaging? In addition I would just use horizontal wind speed, maybe abbreviated as vh as y-label for

(c) figure2: same comments as for figure 1:

We have replaced the labels on Figure 1 and Figure 2 with "Direction [°]" with "$\zeta$ [°]" to be consistent with the later figures and nomenclature used in the text. We deemed that line plots would be more precise for direction in this case since there were little to no instances where the direction changed across the $360°/0°$ transition and the line plots were more precise and more clearly showed measured periodicity. However, to avoid any misinterpretation on the readers' part, we have also changed Figure 1 and Figure 2 to dot plots. Our use of the definition of magnitude for the horizontal wind speed was to minimize any confusion with the velocity components $[U, V, W]$ which were defined earlier in the text. Following the reviewer's suggestion, we have defined $V_h$ and used that for horizontal wind speed. Downsampling was conducted by plotting every 200th data point with no additional anti-aliasing filter applied. We have fixed the caption for Figure 3.

(d) figure 3: again the velocity labels could be much easier and intuitive vh MURC and vh sUAS;

We have changed Figure 3 labels to provide horizontal wind speed as $V_h$.

(e) in addition there is something mixed up in the figure caption: Wind speed is a) and c), not a) and b) as stated in the caption.

This has been corrected.

18. l219: "there is little-to-no evidence of this periodicity! is a rather brave statement; I still see some clear indications of such a periodicity and a formulation

in the direction" shows a distinctly reduced periodicity" sound to me much more appropriate! figure6 and discussion.

The text has been changed as suggested.

19. l221-227;

   (a) what exactly do you want to achieve with this potential temperature profiles? If it is just to give an overview on the state of the atmosphere, then you should distinctly simplify your presentation by e.g. only showing the 4 average profiles (e.g by bin averaging over 25 m vertical intervals;

The authors intended to give an overview on the condition of the atmospheric boundary layer to put the fluctuations of the wind into context. (e.g. fluctuations caused by different turbulence intensity at different altitudes corresponding to the boundary layer stability rather than due to uncorrected bias). However, based on the comments of both reviewers, who saw little value in these profiles, we have removed the potential temperature profiles in the revised manuscript.

   (b) if you are also interested to present an inter-flight variability you can achieve that by using whiskers around your bin mean); as it is it is a rather hard to read/interpret figure figures 7 and 8: those are just messy in the present form; if you want to keep the shown information you could have this as a background with grey and reddish color, but on top you should again show some bin averaged values that would then give a clear picture how mean value and variability/standard deviation react on the proposed correction.

The authors have revised these figures as the reviewer suggested.

20. l235: insert "," before "even"

The authors have added ",".

21. l237: "suggesting a potential time response lag between the five-hole probe and inertial measurement unit"

    (a) insert "the" before "inertial measurement unit"

    The authors have inserted "the" before "inertial measurement unit".

    (b) this has been reported before: Båserud, L., Reuder, J., Jonassen, M. O., Kral, S. T., Paskyabi, M. B., & Lothon, M. (2016). Proof of concept for turbulence measurements with the RPAS SUMO during the BLLAST campaign. Atmospheric Measurement Techniques, 9(10), 4901–4913. https://doi.org/10.5194/amt-9-4901-2016

    The authors have added the following: "which agrees with what was reported in (Båserud et al.,2016)."

22. l244: insert "," after "For Z>200 m"

    The authors have added "," after "For Z>200 m".

23. l256: remove "measured" before "profiles"

    The authors have removed "measured".

24. l264/265: how will a systematic time shift, e.g. introduced by a time delay of the data output of the IMU (as hypothesized in line 237), affect the correction procedure and your results? It might be worth to test (e.g. by a correlation analysis) if there is such a systematic time delay in your data set.

    This particular error is compensated for with the $\Delta t$ term in the correction procedure and is currently one of the larger sources of error requiring correction in our data acquisition systems. We are currently logging IMU/GPS, pressure transducers, and PTU on three separate systems at different rates and time stamps (details provided in Witte et al. 2017). In post-processing we re-align them through cross-correlation of common parameters within each system, but

have found that the automated correlation process can be off (when sampling at $\sim 200$ Hz, even small-time misalignments can cause headaches) so the single optimized time shift was introduced to correct for this misalignment. Hypothetically, the approach could be expanded to account for multiple time shifts which might be introduced due to arrangements, sensor lag, etc. and is something we are currently exploring to correct for strong misalignments introduced during takeoff/landing maneuvers which we believe is due to delays introduced by the EKF calculations made by the IMU.

25. l267: insert "been" after "increasingly"

    The authors have added "been" after "increasingly".

---

## Author Comment (AC3) · 2 Oct 2020

We would like to thank the reviewer for their time spent in review of this manuscript and providing their detailed comments and suggestions. We have prepared a revised manuscript which we believe addresses the concerns and comments raised in their review. In the revised manuscript, specific changes are indicated in blue text, with each of the reviewer's comments addressed below.

1. I wonder why there is no example dataset and example correction code available.

In my opinion, this must be the case.

The reason no data or code have been provided is simply due to the difficulty the authors had in selecting the most appropriate data level to provide.

The easiest approach would be to provide the corrected and uncorrected time series the authors used to produce the figures. However, this level of data would be only suitable for reproducing the figures in the text without allowing the interested party to test the correction themselves. Similarly, the correction described within the paper is contained within five lines of Matlab code centered on the fminsearch command and is of little value without the context of the conversion of raw data to wind measurements.

As the correction is embedded within the processing of the raw data to produce wind vector estimates, the alternative would be providing the original raw data used to produce the time series of wind vector. However, the required data files would consist of raw voltage data from the five-hole probe, pressure, temperature and humidity from the iMet sensor, and six-degree-of-freedom data from the IMU/GPS system. We would then have to provide the numerous subroutines we use to convert the raw voltages to relative velocity, align the data files in time, and extract the wind speed from the relative velocity, as well as provide the individual calibration files for each five-hole-probe and configuration files which are used as part of this process. As well as providing these files, the authors would have to produce sufficient documentation so that an inexperienced user could run the scripts which, ultimately, are custom written for the authors' sUAS measurement system and therefore provide little-to-no value to other sUAS operators who use different hardware.

Instead our goal was to provide a precise description of the correction procedures, with sufficient generality, so that other sUAS operators can implement these procedures within their own data analysis codes.
2. Line 26: Temperature? (e.g. Witte2017)

   Definitely an oversight! The authors have added temperature to the list.

3. Line 32: The difference between the three approaches is not clear: An onboard wind sensor measures air speed, and aircraft kinematics are used to determine ground speed. This seems to be the same as the second approach that you mention. Please briefly explain the differences, it might be helpful to add a reference for each approach (the first approach lacks a reference).

   The authors have revised the sentence as follows: "Wind velocity measurements typically can be partitioned into several approaches: directly by using the instrumentation employing an on-board wind sensor and subtract the aircraft kinematics (Suomi and Vihma, 2018; Cassano et al., 2016); indirectly using the attitude and position data recorded by the inertial measurement unit (IMU) and GPS, respectively (Suomi and Vihma, 2018) ; using both techniques (Rautenberg et al., 2018); or through calibration of the aircraft's kinematic and dynamic response to the wind (González-Rochaet al., 2020)."

4. Line 34: Typo "kinmatic".

   Corrected.

5. Line 36: Sensor-based wind measurements: Isn't everything that measures wind sensor-based? An IMU can be used to determine wind, but it is also a sensor (typically it consists of even 3x3 sensors). Which sensors do you mean?

   The authors have revised the sentence as follows: "Broadly speaking, wind measurements taken by sensors like sonic anemometers, single- and multi-hole pressure probes, and hot-wires tend to have higher temporal (and hence spatial) response."

6. Line 38: Witte2017 writes "Typically, these measurements employ wind velocity probes with a temporal response that is little better than that of sonic anemometers", and "Increasingly, UAVs are utilizing five-hole pressure probes [32,33,40], which can resolve to 40 Hz while flying at approximately 20 m/s.". Today's 3D sonic anemometers can have a data output rate of 100 Hz (e.g. Gill R3-100). So, I am not sure if this is true anymore.

Although such high frequency sonic anemometers do exist, to our knowledge, these anemometers do not exist in a package size sufficiently lightweight to be used on a fixed-wing sUAS. Hot-wire anemometry is also an option, but is not yet in common usage for sUAS measurements and, to our knowledge, multi-sensor hot-wire anemometry capable of resolving the velocity vector has not yet been utilized on sUAS. The authors have removed this sentence as the justification for using multi-hole probes was provided in the sentence following the one described above, and no further justification for their use relative to sonic anemometers is required.

7. Line 142: A change in direction (what direction? Flight direction? Yaw angle?) will result in an acceleration due to a curvature of the flight path. So, what kind of acceleration (rate of change of velocity) do you mean? Flight velocity changes? Vertical acceleration?

We meant the flight direction while the acceleration represents the rate of change in velocity. The authors have revised the sentence to address the reviewer's question as follows: "The assumptions used here are relatively straightforward. The first being that any biasing of $U(t)$ by $U_s$ will result in $U(t)$ having dependence on the direction of travel of the aircraft." and "This portion should not include any significant acceleration or deceleration of the aircraft's horizontal ground speed (e.g. as experienced during takeoff or landing) and should include multiple changes of direction of the aircraft."

8. 146: incomplete sentence

Corrected.

9. Line 194: When you argue with periodicity, then why not show it in a FFT plot?

   The authors have added an FFT plot and corresponding discussion as the reviewer recommended.

10. Line 202: Does $DeltaQ$ have a unit?

    $\Delta Q$ is unitless, it is a correction factor for the dynamic pressure. In retrospect, the authors realized that the $\Delta$ is confusing in this context and have replaced $\Delta Q$ with $\zeta$.

11. Why are figures 5+6 bitmaps and not vector graphs? Is there a way to omit the wrapping-around at $360°$? Is there a better way to convince the readers that the correction improves the accuracy of the data? Because the true velocities are apparently unknown, I would again prefer spectral analyses, that show that the motion of the aircraft becomes less apparent in the corrected data. Color schemes in figures might be better if same colors are used for same objects (e.g. red = sUAS and black = reference). Please also check that colors correctly convert to a gray scale that is distinguishable in black and white print outs.

    The authors have replaced figures 5 and 6 with whiskers plots to better represent the effect of the bias removal.

12. Line 250-266: This seems to be a discussion of the specific weather conditions of that day on that site, I don't see how this adds to the message of the manuscript. Please explain.

    The authors intended to give an overview on the condition of the atmospheric boundary layer to put the fluctuations of the wind into context following correction. (e.g. fluctuations caused by different turbulence intensity at different altitudes corresponding to the boundary layer stability rather than due to uncorrected bias). However, based on the comments of both reviewers who saw little value in

these profiles, we have removed the potential temperature profiles in the revised manuscript.

13. Line 267: "Corrections work": do they improve the data? How do you prove this?

We revised this statement to state that the corrections reduce fluctuations about the mean profile under different conditions for different aircraft. Our justification for concluding that the corrections are working is provided in the final paragraph of the conclusions. Specifically: "Measurements flown near a ground-based reference system revealed significant reduction in measured oscillations of both wind magnitude and direction, which corresponded to the aircraft flight pattern. Additional verification was conducted by comparing profiles of wind speed and direction measured by two different aircraft at two different times. The estimated biases were within $\pm 1°$ for each aircraft, and successful minimization of aircraft-induced oscillations in the measured profiles was observed for both aircraft. These results confirm that the biases are most likely due to physical misalignment of the aircraft and probe axes, as well as demonstrating that the same correction procedures can be applied to multiple aircraft."

14. Line 279: is there a word missing in the first sentence?

Corrected.

---

## Author Response (AR2)

Response to Referee #3

Thank you for your suggestions to improve our manuscript.  Below is the text of the report in blue with our response in black following the relevant comment.  In the marked-up manuscript, these changes are indicated in blue text.

Thanks for implementing the suggested revisions. There is only very minor suggestion for improvement for this revised manuscript:

Figure 5: I think your point would become clearer, if you would only include data from e.g. 9:35 to 9:40. Alternatively, you could scale the y-axis down. There is just too much action per area for my eyes.

We have made some changes to figure 5 to make the graphic clearer.  We have: (1) reduced the horizontal axis range to better illustrate the cyclic nature of the process; (2) switched from a line plot to dot plot to remove lines created when the angle swaps from +180 to -180 degrees; (3) changed the color scheme to reduce the distraction caused by the sUAS orientation cycle.

Line 34: space behind kinematics missing

Space added in

Line 39: space in front of the dot

Space removed

Line 67: Multi-holeS?

Removed superfluous s.

Line 214: Figure reference broken

Fixed figure reference.